# Heme Oxygenase-1 is a Key Molecule Underlying Differential Response of TW-37-Induced Apoptosis in Human Mucoepidermoid Carcinoma Cells

**DOI:** 10.3390/molecules24091700

**Published:** 2019-05-01

**Authors:** In-Hyoung Yang, Chi-Hyun Ahn, Nam-Pyo Cho, Bohwan Jin, WonWoo Lee, Yun Chan Jung, Seong Doo Hong, Ji-Ae Shin, Sung-Dae Cho

**Affiliations:** 1Department of Oral Pathology, School of Dentistry, Institute of Oral Bioscience, Chonbuk National University, Jeonju 54896, Korea; inhyoung3@naver.com (I.-H.Y.); npcho@jbnu.ac.kr (N.-P.C.); 2Department of Oral Pathology, School of Dentistry and Dental Research Institute, Seoul National University, Seoul 03080, Korea; chihyun610@snu.ac.kr (C.-H.A.); hongsd@snu.ac.kr (S.D.H.); 3Laboratory Animal Center, CHA University, CHA Biocomplex, Sampyeong-dong, Seongnam 13488, Korea; jnbhwan@cha.ac.kr (B.J.); lsw232@chamc.co.kr (W.L.); 4Chaon, Seongnamsi 13488, Korea; chaon@chaon.kr

**Keywords:** heme oxygenase-1, TW-37, mucoepidermoid carcinoma, apoptosis, reactive oxygen species

## Abstract

TW-37 is a small-molecule inhibitor of Bcl-2 family proteins, which can induce anti-cancer activities in various types of cancer. In the current study, we investigated the potential molecular mechanism underlying the differential response to TW-37-induced apoptosis in two human mucoepidermoid carcinoma (MEC) cell lines. The differential response and underlying molecular mechanism of human MEC cells to TW-37 was evaluated by trypan blue exclusion assay, western blotting, 4’, 6-diamidino-2-phenylindole staining, annexin V/propidium iodide double staining, analysis of the sub-G1 population, human apoptosis array, and measurements of intracellular reactive oxygen species (ROS). TW-37 decreased cell viability and induced apoptosis in YD-15 cells, but not in MC3 cells. Proteome profiling using a human apoptosis array revealed four candidate proteins and of these, heme oxygenase-1 (HO-1) was mainly related to the differential response to TW-37 of YD-15 and MC3 cells. TW-37 also led to a significant increase in intracellular levels of ROS in YD-15 cells, which is associated with apoptosis induction. The ectopic expression of HO-1 recovered YD-15 cells from TW-37-induced apoptosis by reducing intracellular levels of ROS. The expression of HO-1 was reduced through both transcriptional and post-translational modification during TW-37-mediated apoptosis. We conclude that HO-1 is a potential indicator to estimate response to TW37-induced apoptosis in human MEC.

## 1. Introduction

Salivary gland tumors account for about 0.5% of all malignancies and less than 5% of all head and neck cancers [1]. Mucoepidermoid carcinoma (MEC) is the most common malignant cancer in salivary glands, but there are no effective strategies to manage MEC because of its rarity. Several published studies have demonstrated the anti-tumor activities of single or combination chemotherapy with cisplatin, paclitaxel, or cetuximab [1,2]. We recently observed that MECs are effective in chemotherapy through a variety of molecular mechanisms *in vitro* [3,4]. Despite improvements, current approaches to treating MEC remain disappointing.

The overexpression of anti-apoptotic proteins in the B cell lymphoma-2 (Bcl-2) family, including Bcl-2, Bcl-xL, and myeloid cell leukemia-1 (Mcl-1), can inhibit apoptosis by isolating pro-apoptotic BH3-only proteins [5]. Targeting anti-apoptotic Bcl-2 family proteins is considered an attractive approach to cancer treatment. Thus, small molecule inhibitors targeting anti-apoptotic Bcl-2 family proteins have been developed, and preclinical or clinical trials have attempted to identify the best approach for cancer therapy [6,7,8]. TW-37 is a second-generation benzenesulphonyl derivative of gossypol and is capable of high binding affinity to Bcl-2, Bcl-xL, and Mcl-1 in the nanomolar range [9]. Accumulating studies indicate that TW-37 displays anti-proliferative or pro-apoptotic properties in various types of cancers by inhibiting Bcl-2 and/or Mcl-1 *in vitro* and *in vivo* [10,11]. TW-37 also inhibits tumor growth and induces apoptosis by targeting alternative signaling molecules such as notch-1, Jagged-1, or prostate apoptosis response-4 without affecting Bcl-2 family members [12,13]. On the other hand, deficiencies of Bax or Bak and the overexpression of neuroglobin mutation protects cells from TW-37-mediated cell death, indicating that biological conditions may control TW-37-induced apoptosis [14,15]. Considering this evidence, the identification of a molecular target for TW37-induced apoptosis is needed.

In the present study, we sought a novel molecular target for TW-37 to effectively induce antitumor activity in MEC cell lines, irrespective of anti-apoptotic Bcl-2 family proteins.

## 2. Results

### 2.1. Comparative Analysis of Anti-Proliferative and Pro-Apoptotic Effects of TW-37 in Human MEC Cell Lines

To investigate the anti-proliferative effects of TW-37 in two human MEC cell lines (MC3 and YD-15) *in vitro*, both cell lines were treated with vehicle control (DMSO) or 1.25 μM TW-37 for the indicated time points. TW-37 resulted in significant decreases in cell viability in YD-15 cells (Figure 1A). The IC50 values for inhibition of cell viability at 24 and 48 h were 1.41 and 0.85 μM, respectively. In contrast, no growth inhibitory effects of TW-37 at the same concentration were observed in MC3 cells. To determine whether the anti-proliferative effects of TW-37 were related to the induction of an apoptotic response, we examined levels of PARP cleavage, a hallmark of apoptosis. Treatment of TW-37 for 48 h clearly induced the cleavage of PARP in YD-15 cells but not in MC3 cells (Figure 1B). Although 24 h treatment of TW-37 in YD15 cells decreased cell viability, it induced s-phase cell cycle arrest instead of an increase in the cleavage of PARP (Appendix A). Consistent with these observations, fluorescence-based results from DAPI staining showed that TW-37 led to obvious induction of nuclear condensation and DNA fragmentation, typical features of apoptosis (Figure 1C). Then, we confirmed TW-37-induced apoptosis by assessing flow cytometry experiments. The rate of annexin V-positive cells in YD-15 cells was increased from 12.78% of the control group to 19.42% of the TW-37 treatment group (Figure 1D). The proportion of sub-G1 phase YD-15 cells in the TW-37 treatment group was 7.30%, compared with 1.36% of control group (Figure 1E). However, no significant changes in induction of apoptosis were observed in MC3 cells. These results indicate that YD-15 cells were sensitive to inhibition of cell growth and induction of apoptosis under TW-37 treatment, unlike MC3 cells.

### 2.2. Target Profiling Related to Differential Response of Human MEC Cells to TW-37

To identify the causes of differential response to TW-37 between MC3 and YD-15 cells, we evaluated the relative expression levels of 35 apoptosis-related proteins using a human apoptosis array. Four proteins (HO-1, HSP27, cleaved caspase 3, and DR5) were changed by TW-37 treatment in YD-15 cells compared with in MC3 cells (Figure 2A,B). To verify results from the human apoptosis array, the levels of four proteins were detected by western blotting. TW-37 resulted in pronounced reductions of anti-apoptotic proteins (HO-1 and HSP27), whereas the levels of pro-apoptotic proteins (cleaved caspase 3 and DR5) were increased in YD-15 cells (Figure 2C). However, no significant differences in protein levels were observed in MC3 cells. These results indicate that these four proteins affect the response of two human MEC cell lines to TW-37.

### 2.3. Role of HO-1 in Oxidative Stress-Induced Apoptosis Upon TW-37 Treatment

To clarify the biological significance of HO-1 or HSP27 in TW-37-induced apoptosis, YD-15 cells were transiently transfected with pcDNA3.1 (a control vector) and pcDNA3.1-HO-1 (to overexpress HO-1) or pcDNA3.1-HSP27 (to overexpress HSP27). The ectopic expression of HO-1 significantly recovered YD-15 cells from TW-37-induced PARP cleavage (Figure 3A). However, no significant changes in PARP cleavage were observed in HSP27-overexpressing cells (Appendix A). To confirm whether the ectopic expression of HO-1 could attenuate apoptosis induced by TW-37, we assessed a combinational assay using annexin V/PI staining. The rate of apoptotic cells was lower in HO-1-overexpressing cells following TW-37 treatment (23.05%) compared with the control group (28.08%) (Figure 3B). Consistent with these results, the proportion of sub-G1 phase in HO-1-overexpressing cells following TW-37 treatment was 17.94% compared with 26.61% of the control group (Figure 3C).

### 2.4. Role of Intracellular ROS Generation in TW-37-Induced Apoptosis

To elucidate whether the sensitive response of TW-37 to apoptosis in YD-15 cells is potentially involved in oxidative stress, we detected intracellular levels of ROS using H2DCF-DA fluorescent dye. TW-37 led to pronounced increase in intracellular ROS production, which was abolished by a ROS scavenger, N-acetyl cysteine (NAC) (Figure 4A). To confirm whether the ROS generation induced by TW-37 contributed to apoptosis in YD-15 cells, we performed fluorescence-based analysis. The typical morphological features of apoptotic cells induced by TW-37 were in part abrogated by NAC co-treatment (Figure 4B). Consistent with these results, the results of flow cytometry analysis indicated that NAC partly led to a decrease in apoptotic cells in the presence of TW-37 (Figure 4C,D). To further clarify the correlation between the two proteins (HO-1 and HSP27) and oxidative stress in TW-37-induced apoptosis, western blotting analysis was employed. Co-treatment with TW-37 and NAC in part abolished the cleavage of PARP compared with TW-37 treatment alone, but no significant changes in levels of HO-1 and HSP27 were observed (Figure 4E), suggesting that HO-1 or HSP27 influences intracellular ROS generation-mediated apoptosis. Next, we further investigated the biological role of HO-1 in oxidative stress-induced apoptosis upon TW-37 treatment. The ectopic expression of HO-1 in part decreased the intracellular ROS production induced by TW-37 (Appendix A). These results suggest that HO-1 plays an important role in oxidative stress-induced apoptosis upon TW-37 treatment, and that intracellular ROS generation is required for TW-37-induced apoptosis in YD-15 cells, which might be accompanied by regulation of HO-1 expression.

### 2.5. Transcriptional and Post-Translational Regulation of HO-1 by TW-37

In order to clarify how HO-1 protein was reduced by TW-37, we first analyzed the expression levels of *HO-1* mRNA using qPCR. TW-37 treatment decreased *HO-1* mRNA levels at two time points (12 h and 24 h), whereas after reduction by TW-37 treatment for 48 h, the levels returned to the original state (Figure 5A). Previously, Lin et al. [16] suggested that protein turnover was involved in the regulation of HO-1 protein expression. To address this question, the effect of TW-37 on HO-1 protein turnover in YD-15 cells was analyzed using cycloheximide (CHX), a new protein synthesis inhibitor. Co-treatment with TW-37 and CHX reduced HO-1 protein to a greater degree than did single treatment with TW-37 (Figure 5B). To investigate the relationships of proteasomes with TW-37-induced HO-1 protein degradation, we assayed the effect of the proteasome inhibitor MG132 on HO-1 and found that the decrease in HO-1 protein level by TW-37 was significantly reversed by MG132 (Figure 5C). Collectively, these results suggest that HO-1 is modulated by TW-37 through both transcriptional and post translational modifications. 

## 3. Discussion

In this report, we unexpectedly found that TW-37 showed anti-proliferative and pro-apoptotic activities in YD-15 cells, but not MC3 cells. Based on these data, we used a proteome profiling approach to identify a novel molecular target that is responsible for the differential responses of two MEC cell lines to TW-37. We found that the new molecular mechanism may depend on heme oxygenase-1 (HO-1) expression. HO-1 catalyzes the oxidative degradation of cellular heme to release free iron, carbon monoxide, and biliverdin in mammalian cells, and is strongly induced by several cellular stresses including ROS, heavy metal, ultraviolet light, and hypoxia [17]. HO-1 is expressed at low levels under basal conditions, but is up-regulated as a defense mechanism in response to a variety of pathological stresses [18]. Interestingly, increases in HO-1 expression have been observed in various types of cancer including primary chronic myeloid leukemia cells [19], pancreatic cancer cells [20], and murine melanoma [21]. HO-1 is also linked to metastasis in oral squamous cell carcinoma [22] and promotes angiogenesis through VEGF in breast cancer cells [23]. Other prior studies demonstrated that inhibition of HO-1 resulted in increased chemo-sensitivity of cancer cells to chemotherapeutic drugs [20,22,24]. Thus, HO-1 may serve as an essential modulator for the initiation and progression of human cancer. TW-37 effectively led to a decrease in HO-1 expression in YD-15 cells and ectopic expression of HO-1 led to recovery of YD-15 cells from TW-37-induced apoptotic activity. The most obvious finding is that differences in apoptotic responsiveness to TW-37 in two human MEC cell lines are mainly associated with the expression levels of HO-1 protein.

HO-1 functions as an antioxidant enzyme to maintain redox homeostasis due to its byproducts [25]. As an important enzyme in oxidative stress response, HO-1 protects cells, resists apoptosis, and induces drug resistance [26,27]. On the other hand, HO-1 degradation by TRC8 E3 ligase promotes ROS generation-induced DNA damage by triggering mitotic delay at G2/M phase cell cycle arrest [28]. It is therefore likely that such connections exist between HO-1 and ROS. To test this hypothesis, we investigated the involvement of oxidative stress in TW-37 induced apoptosis in MEC cell lines. The intracellular levels of ROS in YD-15 cells increased upon TW-37 treatment, which contributed to TW37-induced apoptosis. To improve our understanding of the potential role of HO-1 in ROS-induced apoptosis, we examined the effect of ectopic expression of HO-1 on ROS generation during the apoptosis of YD-15 cells. As expected, ectopic expression of HO-1 partially abrogated ROS generation-mediated apoptosis induced by TW-37. Collectively, the differential apoptotic responsiveness to TW-37 may be related to HO-1-mediated ROS generation in human MEC cell lines.

Heat shock protein 27 (HSP27) is known to play a major role as a protein chaperone and antioxidant, and is related to poor prognosis [29]. In this study, we also observed that HSP27 was reduced in YD-15 cells upon TW-37 treatment. This finding prompted further investigations of the biological role of HSP27 in TW-37-induced apoptosis. However, ectopic expression of HSP27 failed to recover YD-15 cells from PARP cleavage upon TW-37 treatment, implying that HSP27 is not essential for TW-37-induced apoptosis, unlike HO-1. In a previous study, knockdown of HSP27 by siRNA technology increased paclitaxel-induced ROS production and chemo-sensitivity [30]. It is possible that HSP27 is involved in TW-37-generated ROS. We also observed that TW-37 led to a pronounced increase in death receptor 5 (DR5) expression in YD-15 cells, but not in MC3 cells. ROS is considered a key regulator of DR5-dependent apoptosis. Previous studies demonstrated that the accumulation of intracellular ROS by anti-cancer agents facilitated upregulation of DR5 expression to induce extrinsic apoptotic pathways [31,32]. Several chemicals sensitize cancer cells to TRAIL-mediated apoptosis via ROS generation-dependent endoplasmic reticulum (ER) stress that in turn upregulates DR5 expression [33,34]. We found that TW-37 resulted in a significant increase of C/EBP homologous protein (CHOP), as known as ER stress marker protein (data not shown). Thus, we hypothesize that ROS accumulation by TW-37 may contribute to ER stress-mediated upregulation of DR5 during apoptosis in YD-15 cells. However, the role of DR5 expression remains incompletely understood, and further research should examine more closely the links between DR5 and differential response to TW-37 of two human MEC cell lines.

## 4. Materials and Methods

### 4.1. Chemicals and Antibodies

TW-37 was purchased from ApexBio (Houston, TX, USA) and dissolved in dimethyl sulfoxide (DMSO), aliquoted, and stored at −20 ℃. We acquired 4’,6-diamidino-2-phenylindole (DAPI), propidium iodide (PI), N-acetyl-L-cysteine (NAC), and cycloheximide from Sigma-Aldrich (Saint Louis, MO, USA). Annexin V-FITC/PI was supplied by BD Biosciences (Franklin Lakes, NJ, USA) and 2ʹ,7ʹ-dichlorofluorescin diacetate (H2DCF-DA) was obtained from Calbiochem (San Diego, CA, USA). Antibodies against cleaved PARP, cleaved CASPASE 3, HSP27, and DR5 were supplied by Cell Signaling Technology, Inc. (Charlottesville, VA, USA). MG-132 and antibodies against HO-1 and β-ACTIN were obtained from Santa Cruz Biotechnology, Inc. (Dallas, TX, USA).

### 4.2. Cell Culture and Treatment

MC3 and YD-15 cell lines (human mucoepidermoid carcinoma) were provided by Fourth Military Medical University (Xi’an, China) and Yonsei University (Seoul, Republic of Korea), respectively. Both cell lines were grown in either DMEM/F12 or RPMI1640 media supplemented with 10% fetal bovine serum and 1% penicillin/streptomycin at 37 ℃ in a 5% CO2 incubator. When the cells reached 40% confluence, they were treated with DMSO or 1.25 μM of TW-37. For all experiments, both adherent and floating cells were collected.

### 4.3. Trypan Blue Exclusion Assay

Cells were seeded in 6-well plates and incubated overnight prior to treatment with TW-37. After treatment, cells were stained with 0.4% trypan blue solution (Gibco, Paisley, UK) and viable cells were counted using a hemocytometer.

### 4.4. Western Blotting

Cells were extracted with RIPA lysis buffer (EMD Millipore, Billerica, CA, USA) using phosphatase inhibitor and protease inhibitor cocktails. Protein quantification was performed using a DC protein assay kit (BIO-RAD Laboratories, Madison, WI, USA). After normalization, protein lysates containing approximately 20–50 μg of protein were boiled with protein sample buffer at 95 ℃ for 5 min and separated on SDS-PAGE. After electrophoresis, the proteins were transferred to immuno-blot polyvinylidene difluoride membranes and blocked with 5% skim milk for 1 h at room temperature (RT). The membranes were incubated with the indicated primary antibodies overnight at 4 ℃, and maintained with corresponding horseradish peroxidase (HRP)-conjugated secondary antibodies for 2 h at RT. The bands were immune-reactivated with ECL solution (Santa Cruz Biotechnology, Inc., Santa Cruz, CA, USA) and visualized by ImageQuant LAS 500 (GE Healthcare Life Sciences, Piscataway, NJ, USA) or X-ray film.

### 4.5. DAPI Staining

After treatment, cells were fixed with 100% ethanol at 20 ℃ overnight, deposited on slides, and stained with DAPI solution (2 μg/ml). Morphological changes of nuclei in apoptotic cells were observed using a fluorescence microscope (Leica DMi8, Wetzlar, Germany).

### 4.6. Annexin V/PI Double Staining

The induction of apoptosis was measured using an FITC Annexin V Apoptosis Detection Kit (BD Pharmingen, San Jose, CA, USA) according to the manufacturer’s protocol. Briefly, floating and adherent cells were collected, washed twice with PBS, and pelleted by centrifugation. Then, cells were resuspended in annexin V binding buffer containing 3 μl annexin V-FITC and 1 μl PI, and incubated for 15 min at RT in the dark. Subsequently, cells were transferred to a FACS tube and analyzed by flow cytometry using a FACSCalibur (BD Biosciences, San Jose, CA, USA), and at least 10,000 events were counted per sample.

### 4.7. Analysis of the Sub-G1 Population

In order to measure the numbers of cells in the sub-G1 phase, floating and adherent cells were collected and fixed in 70% ethanol overnight at −20 ℃. After washing with PBS, the cells were incubated with propidium iodide solution (20 μg/ml) and RNase A (20 μg/ml) for 15 min at 37 ℃. The cell cycle distribution, including the sub-G1 phase, was analyzed by FACSCalibur and at least 10,000 events were counted per sample.

### 4.8. Human Apoptosis Array

To evaluate the relative expressions of 35 apoptosis-related proteins, we used the Human Apoptosis Array Kit (R&D Systems, Minneapolis, MN, USA). Briefly, nitrocellulose membranes were blocked with an array buffer for 1 h at RT. Protein lysates were diluted, added, and incubated overnight. After washing with 1X wash buffer to remove unbound proteins, membranes were exposed to a cocktail of biotinylated detection antibodies for 1 h at RT. Membranes were then washed and incubated with streptavidin–HRP for 30 min at RT. Each capture spot corresponding to the amount of apoptotic bound protein was detected with enhanced chemiluminescence western blotting luminol reagent and visualized by X-ray film.

### 4.9. Measurement of Intracellular ROS

ROS measurements were performed using a cell-permeable fluorogenic probe, H2DCF-DA. Briefly, cells were seeded on 4-well plates and treated with DMSO or TW-37 in the absence or presence of NAC. After 48 h, cells were washed twice with PBS. We added 20 µM of H2DCF-DA and incubated the mixture for 30 min at 37 ℃ in a CO2 incubator. Cells were then observed under a fluorescence microscope.

### 4.10. Construction of Overexpression Vectors and Transient Transfection

The open reading frame of human HO-1 and HSP27 was amplified from cDNA that was synthesized in YD-15 cells using gene-specific primers (primer sequence; *HO-1* sense, 5′-GAA TTC ATG GAG CGT CCG CAA CCC GAC AGC-3′, with an included EcoRI site, *HO-1* anti-sense, 5′-GAA TTC TCA CAT GGC ATA AAG CCC TAC AGC AAC TG-3′, with an included EcoRI site, *HSP27* sense, 5′-GAA TTC ATG ACC GAG CGC CGC GTC CCC TTC-3′, with an included EcoRI site, *HSP27* anti-sense, 5′-GAA TTC TTA CTT GGC GGC AGT CTC ATC GGA T-3′, with an included EcoRI site), and then cloned into pGEM^®^ T-easy vector (Promega, Madison, WI, USA). The *HO-1* and *HSP27* were confirmed by sequence analysis. Finally, the genes were cloned into the multi cloning site of the pcDNA3.1 (+) vector (Invitrogen, San Diego, CA, USA). For overexpression vector transfection, cells were transfected with vector constructs (pcDNA3.1; pcDNA3.1-HO-1 or pcDNA3.1-HSP27) using Lipofectamine 3000 transfection reagent (Life Technologies, Carlsbad, CA, USA) according to the manufacturer’s instructions.

### 4.11. Quantitative Real-Time PCR

Total RNA was extracted using Trizol Reagent (Life Technologies, Carlsbad, CA, USA). One microgram of RNA was reverse-transcribed by an AMPIGENE cDNA Synthesis Kit (Enzo Life Sciences, Inc., NY, USA), and the resultant cDNA was subjected to PCR using AMPIGENE qPCR Green Mix Hi-Rox (Enzo Life Sciences, Inc.). Real-time PCR was performed using the Applied Biosystems StepOne Plus Real-Time PCR System (Applied Biosystems, CA, USA) and PCR conditions for all genes were as follows: 95 ℃ for 2 min, followed by 40 cycles of 95 ℃ for 10 sec and 60 ℃ for 30 sec. The relative amount of each gene was normalized to the amount of *GAPDH* and calculated using the 2-ΔΔCt method. The qPCR primers were: *HO-1* sense, 5′-CCA GCA ACA AAG TGC AAG AAT C-3′, *HO-1* anti-sense, 5′-CCA CCA GAA AGC TGA GTG TAA G-3′, *GAPDH* sense, 5′-GTG GTC TCC TCT GAC TTC AAC-3′, *GAPDH* anti-sense, 5′-CCT GTT GCT GTA GCC AAA TTC-3′.

### 4.12. Statistical Analysis

Statistical analyses were performed using SPSS version 22 (SPSS Inc., Chicago, IL, USA). Two-tailed Student’s *t*-tests were used to compare two experiments, and one-way ANOVA for multiple comparisons with Tukey’s post hoc test. Statistical significance was set at *p* < 0.05.

## 5. Conclusions

In this study, we provide the first mechanistic evidence that inhibition of HO-1 may determine response to TW-37 in human MEC cell lines. Thus, these findings reveal a potential strategy to improve response to TW-37 for the treatment of human MEC.

## Figures and Tables

**Figure 1 molecules-24-01700-f001:**
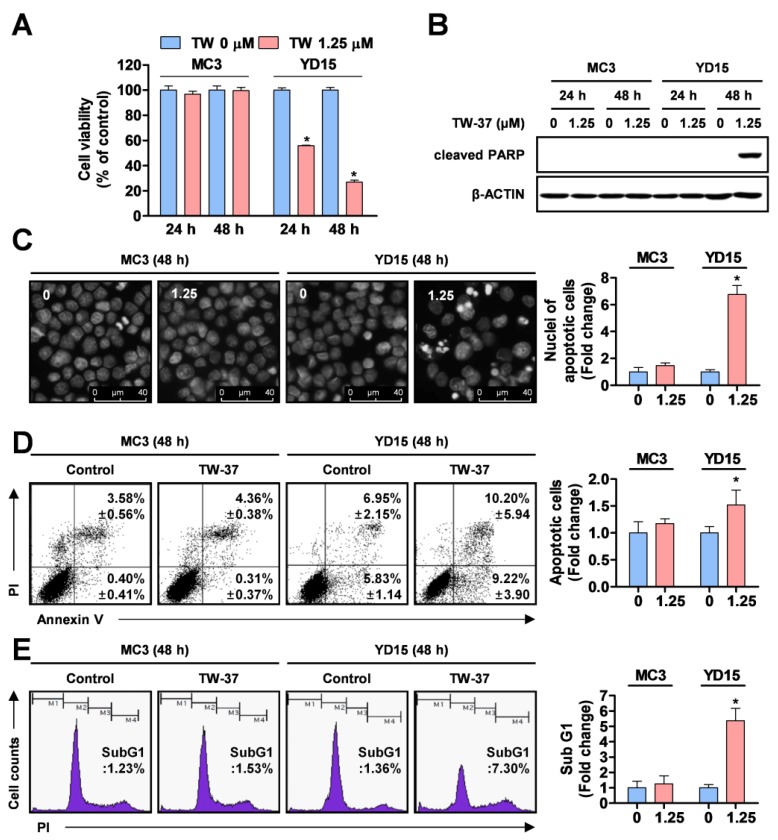
Effects of TW-37 on cell viability and apoptosis of human mucoepidermoid carcinoma (MEC) cell lines. MC3 and YD-15 cells were treated with dimethyl sulfoxide (DMSO) or a 1.25 μM concentration of TW-37 for 24 h or 48 h. (**A**) Cell viability was examined by a trypan blue exclusion assay. The results represent the mean ± SD of triplicate from three independent experiments and significant differences are shown (*, *p* <0.05). (**B**) Cleavage of poly (ADP-ribose) polymerase (PARP) was analyzed by western blotting and actin was used as a loading control. (**C**) Nuclear fragmentation and DNA condensation were visualized by fluorescence microscopy. Representative images from three independent experiments are shown (magnification, X400). (**D**) Fluorescence-activated cell sorting (FACS) analysis of annexin V/ propidium iodide (PI) staining. The annexin V-positive cells are expressed as the percentage of apoptotic cells. (**E**) FACS analysis of PI staining. The histogram represents the distribution of cell cycles and the graphs represent the mean ± SD from three independent experiments. *, *p* < 0.05.

**Figure 2 molecules-24-01700-f002:**
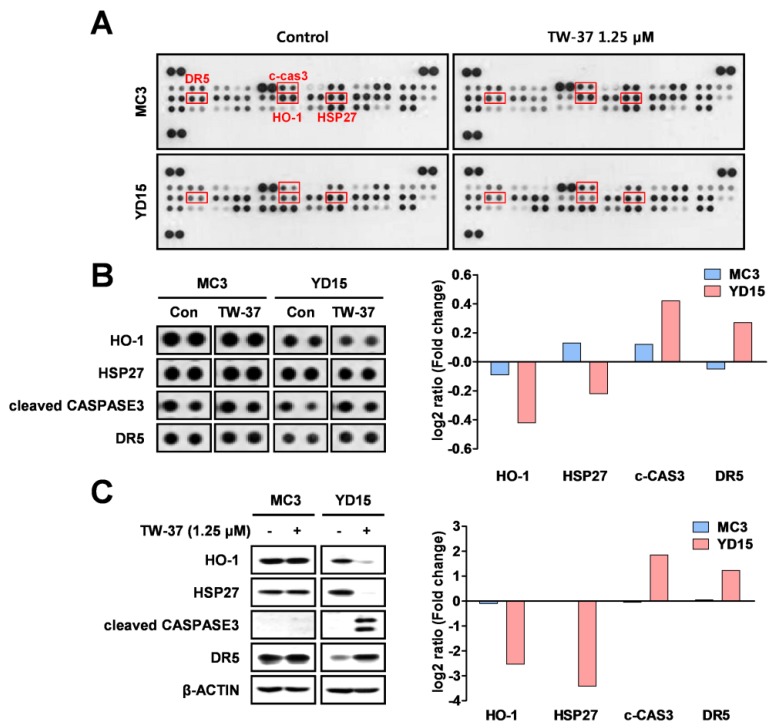
Target profiling of TW-37 in human MEC cell lines. MC3 and YD-15 cells were treated with DMSO or a 1.25 μM concentration of TW-37 for 48 h. (**A**) Entire images of the human apoptosis array. (**B**) Differential expression levels of four candidate proteins are shown. The differences in expression levels of four proteins were analyzed using Image J software. (**C**) Target verification of results from the human apoptosis array by western blotting. Actin was used as a loading control.

**Figure 3 molecules-24-01700-f003:**
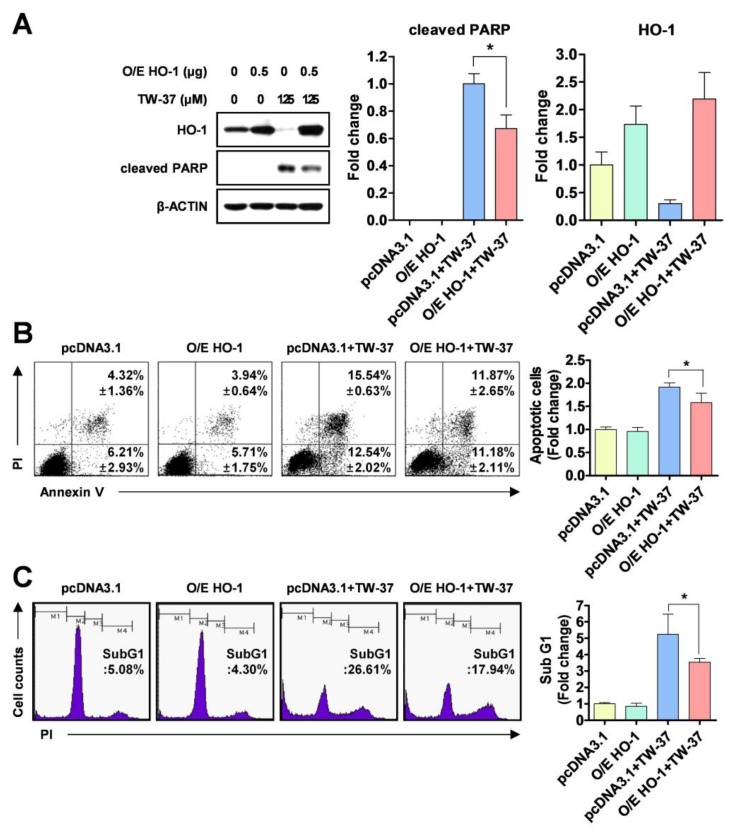
Blockade of TW-37-induced apoptosis by ectopic expression of HO-1. YD-15 cells were transiently transfected with pcDNA3.1 or pcDNA3.1-HO-1 for 6 h, and then treated with DMSO or TW-37 for 48 h. (**A**) Protein levels of HO-1 and cleaved PARP were analyzed by western blotting. Actin was used as a loading control. Bar graphs are expressed as mean ± SD from three independent experiments. *, *p* < 0.05. (**B**) FACS analysis of annexin V/PI staining. The annexin V-positive cells are expressed as the percentage of apoptotic cells. The results are expressed as mean ± SD from three independent experiments. *, *p* < 0.05. (**C**) FACS analysis of PI staining. The histogram represents the distribution of the cell cycle and the percentage of cells in the sub-G1 phase are expressed as mean ± SD from three independent experiments. *, *p* < 0.05.

**Figure 4 molecules-24-01700-f004:**
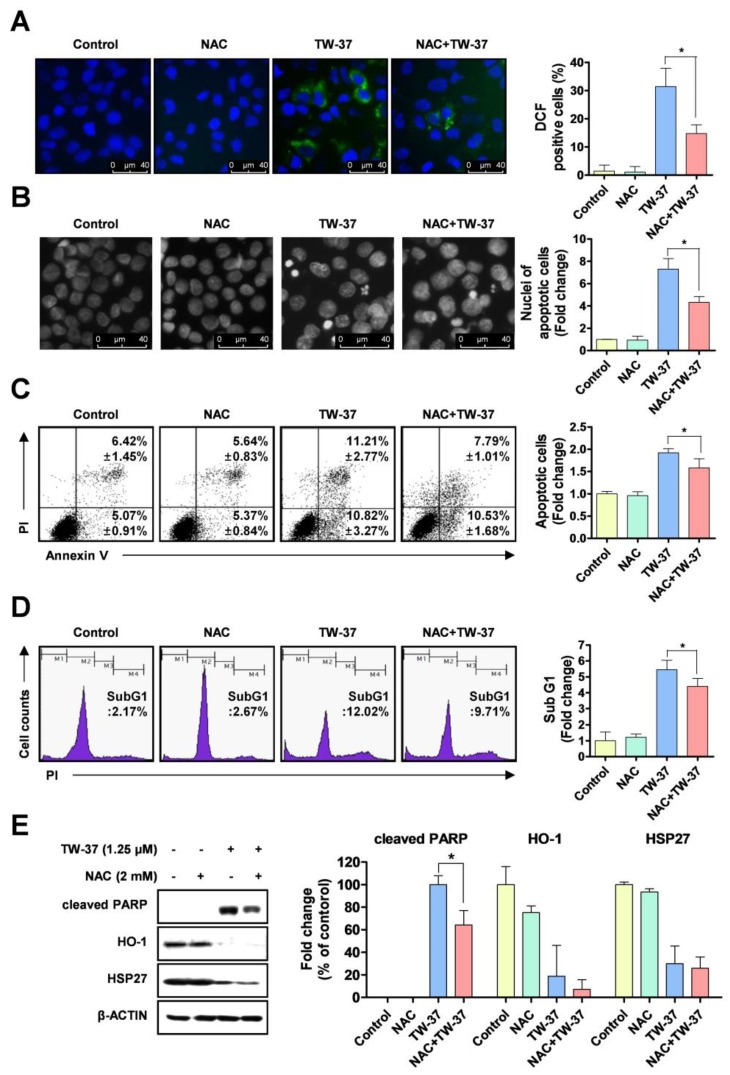
Role of reactive oxygen species (ROS) generation in TW-37-induced apoptosis. YD-15 cells were treated with DMSO or TW-37 in the absence or presence of N-acetyl cysteine (NAC) for 48 h. (**A**) Intracellular levels of ROS in the cells were detected using H2DCF-DA fluorescent dye and visualized by fluorescence microscope (magnification, X400). (**B**) Fluorescence microscope analysis of DAPI staining. Representative images from three independent experiments are shown (magnification, X400) and the graphs are expressed as mean ± SD. *, *p* < 0.05. (**C**) FACS analysis of annexin V/PI staining. The annexin V-positive cells are expressed as the percentage of apoptotic cells. Bar graphs are expressed as mean ± SD from three independent experiments. *, *p* < 0.05. (**D**) FACS analysis of PI staining. The histogram represents the distribution of the cell cycle and the percentage of cells in the sub-G1 phase are expressed as mean ± SD from three independent experiments. *, *p* < 0.05. (**E**) Protein levels of cleaved PARP, HO-1, and HSP27 were assessed by western blotting. Actin was used as a loading control. Bar graphs represent the mean ± SD of three independent experiments. *, *p* < 0.05.

**Figure 5 molecules-24-01700-f005:**
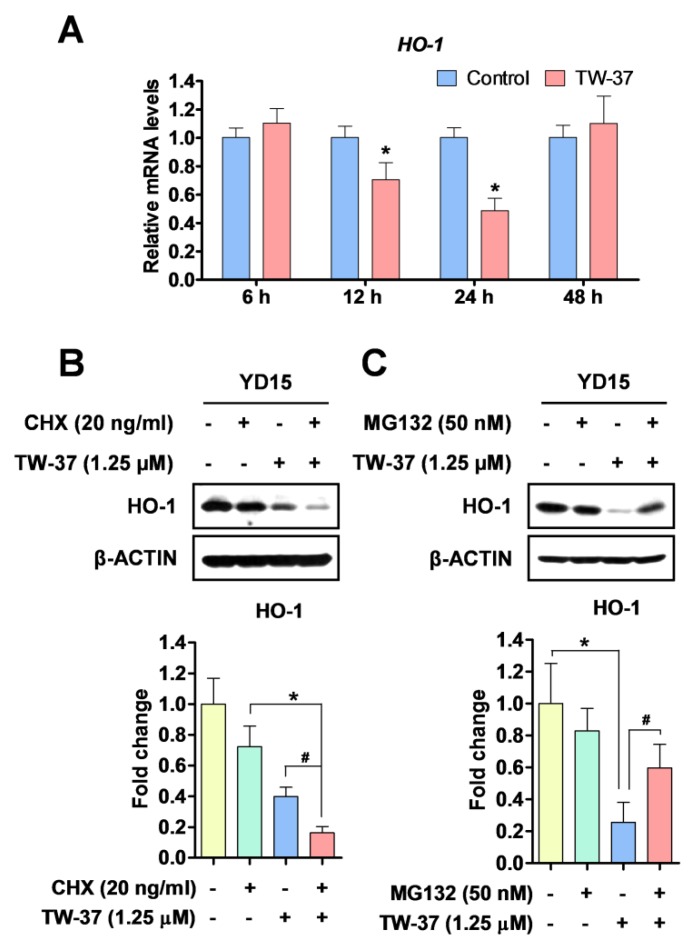
The regulation of HO-1 protein in TW-37-induced apoptosis in YD-15 cells. (**A**) YD-15 cells were treated with DMSO or TW-37 and harvested at the indicated time points (6, 12, 24, and 48 h). *HO-1* expression was examined by qPCR and expression was normalized to *GAPDH*. YD-15 cells were pretreated for 1 h with CHX (**B**) or MG132 (**C**). After 1 h pretreatment, TW-37 or DMSO were added for 24 h. Protein levels of HO-1 was analyzed by western blotting. Bar graphs represent the mean ± SD of three independent experiments. *, *p* < 0.05 compared with the DMSO-treated group. ^#^, *p* < 0.05 compared with the TW-37-treated group.

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
