# Peer review of "Heme Oxygenase-1 is a Key Molecule Underlying Differential Response of TW-37-Induced Apoptosis in Human Mucoepidermoid Carcinoma Cells"

_molecules, 2019, doi:10.3390/molecules24091700_

Round 1
Reviewer 1 Report
The manuscript demonstrates the crucial role of heme oxygenase 1 in preventing apoptosis induced by TW-37, a small-molecule inhibitor of Bcl-2 family proteins. Being aware of research aimed at inhibition of this enzyme for the sake of cancer therapy, I find this study very interesting, instructive and useful. The experiments were well planned, correctly performed, and properly described.
Minor remarks:
Fig. 3A. The blot shown does not seem too representative for the histogram (compare O/E HO-1 and O/E HO-1 + TW37).
Fig. 1D, 3B, 4C. Bar graphs are expressed as mean ± SD from three 169 independent experiments. Do values on the scattergrams represent also mean values or values for a single representative measurement?
Discussion: Do the authors have any ide how HSP27 could be involved in the suppression of ROS generation?
Author Response
Reviewer #1:
The manuscript demonstrates the crucial role of heme oxygenase 1 in preventing apoptosis induced by TW-37, a small-molecule inhibitor of Bcl-2 family proteins. Being aware of research aimed at inhibition of this enzyme for the sake of cancer therapy, I find this study very interesting, instructive and useful. The experiments were well planned, correctly performed, and properly described.
Minor remarks:
Fig. 3A. The blot shown does not seem too representative for the histogram (compare O/E HO-1 and O/E HO-1 + TW37).
☞ Thank you for your kind and valuable comments. As you suggested, we did an additional experiment and we re-quantified our three independent experiments of western blotting. The below is the change of our Fig. 3A. We added new Fig. 3. [line 133]
Fig. 3A.
Fig. 1D, 3B, 4C. Bar graphs are expressed as mean ± SD from three independent experiments. Do values on the scattergrams represent also mean values or values for a single representative measurement?
☞ Values on the scattergrams represent mean values of three independent experiments. The below is our correction. We added new Fig. 1, 3 and 4. [lines 91, 133, and 164]
Discussion: Do the authors have any idea how HSP27 could be involved in the suppression of ROS generation?
☞ Thank you for your valuable comments. There are other several reports showing that HSP27 could be involved in the suppression of ROS generation. HSP27 can have protective activity against oxidative stress by upholding glutathione in its reduced form and by decreasing iron intracellular levels (Ref. 1). The active form of HSP27 can control intracellular reactive oxygen species and glutathione levels (Ref. 2). These reports supported that HSP27 is deeply related to the suppression of ROS.
Ref 1 Arrigo AP1, Virot S, Chaufour S, Firdaus W, Kretz-Remy C, Diaz-Latoud C. Hsp27 consolidates intracellular redox homeostasis by upholding glutathione in its reduced form and by decreasing iron intracellular levels. Antioxid Redox Signal. 2005 Mar-Apr;7(3-4):414-22.
Ref. 2 Mehlen P1, Hickey E, Weber LA, Arrigo AP. Large unphosphorylated aggregates as the active form of hsp27 which controls intracellular reactive oxygen species and glutathione levels and generates a protection against TNFalpha in NIH-3T3-ras cells. Biochem Biophys Res Commun. 1997 Dec 8;241(1):187-92

Reviewer 2 Report
Mucoepidermoid carcinomas (MECs) are rare, however, they display a severe health burden. Thus, it is essential to elucidate underlying molecular mechanisms in order to develop therapeutic strategies. In their study, the authors utilized two human MEC cell lines, MC3 and YD-15, to evaluate the effect of TW-37 which inhibits proteins of the BCL2 family. For this purpose, they have used a several approaches to assess apoptosis-related effects on these cell lines. The impact of this study is clearly limited because only one TW-37-responsive and one TW-37-unresponsive cell line was used. However, their findings on this topic indicate the role of HO-1 in the efficiency of TW-37 in MEC cell in vitro and will be of interest to the readership of Molecules. A number of major and minor points should be addressed prior to publication as detailed below.
In this study only one TW-37-responsive and one TW-37-unresponsive cell line was used. Therefore, it is possible that the effects observed here are cell line-specific. To support the findings of the relevance of HO-1 in the TW-37-associated response, another TW-37-responsive and TW-37-unresponsive cell line should be included.
The authors show that YD15 cells were sensitive to TW-37 treatment while MC3 cells were not. This was, among others shown using a cell viability assay of cells treated with 0 µM and 1.25 µM TW-37. Because the response of different cell lines towards specific inhibitors can be highly heterogeneous, it is possible, that higher concentrations are required for MC3 cells. Please show cell viability and cell morphology (conventional light microscopy) using a wide range of TW-37 concentrations.
The flow cytometry profiles shown in Figure 1D suggest that the TW-37-treated MC3 cells were not gated correctly.
Please show mRNA levels of HO-1 and HSP27 after overexpression using qRT-PCR.
The H2DCF-DA staining (Figure 4A) seems to be unspecific (high background staining). Please provide pictures with a higher magnification which make clear that the staining was specific.
Please indicate for all experiments whether only adherent or also floating cells were collected.
Cleaved PARP is not expressed (suggesting that it results from gene transcription) as mentioned in the text (line 95) but results after cleavage of the PARP protein by active caspases. Please avoid the term “expression” in this context.
Please add scale bars to all microscopy images.
Please italicize “in vivo” and “in vitro”.
Human gene names should be written in capital letters and be italicized. Human proteins are written in capital letters. Murine gene names should be written in small letters (only the first letter is capitalized) and be italicized. Murine proteins are written in capital letters.
Author Response
Reviewer #2:
Mucoepidermoid carcinomas (MECs) are rare, however, they display a severe health burden. Thus, it is essential to elucidate underlying molecular mechanisms in order to develop therapeutic strategies. In their study, the authors utilized two human MEC cell lines, MC3 and YD-15, to evaluate the effect of TW-37 which inhibits proteins of the BCL2 family. For this purpose, they have used a several approaches to assess apoptosis-related effects on these cell lines. The impact of this study is clearly limited because only one TW-37-responsive and one TW-37-unresponsive cell line was used. However, their findings on this topic indicate the role of HO-1 in the efficiency of TW-37 in MEC cell in vitro and will be of interest to the readership of Molecules. A number of major and minor points should be addressed prior to publication as detailed below.
In this study only one TW-37-responsive and one TW-37-unresponsive cell line was used. Therefore, it is possible that the effects observed here are cell line-specific. To support the findings of the relevance of HO-1 in the TW-37-associated response, another TW-37-responsive and TW-37-unresponsive cell line should be included.
☞ Thank you for your kind and valuable comments. These are really good and important comments to us. In this study, we focused on mucoepidermoid carcinoma of salivary gland. For the study on MECs, only a few cell lines are available and we have only two different cell lines, MC3 and YD-15. Thus, please, understand our lab situations. Previously, Kongpetch S. et al. (2012) and Ma D. et al. (2014) demonstrated crucial role of HO-1 on the sensitivity of cholangiocarcinoma cells or acute myeloid leukemia cells to chemotherapeutic agents (Ref.1 and Ref.2). These suggest that HO-1 could be deeply involved in the sensitivity of cancer cells to anti-cancer drugs. However, we should describe that the effects observed here are MEC cell line-specific as you commented.
Ref.1 Kongpetch S1, Kukongviriyapan V, Prawan A, Senggunprai L, Kukongviriyapan U, Buranrat B. Crucial role of heme oxygenase-1 on the sensitivity of cholangiocarcinoma cells to chemotherapeutic agents. PLoS One. 2012;7(4):e34994.
Ref.2 Ma D1, Fang Q, Li Y, Wang J, Sun J, Zhang Y, Hu X, Wang P, Zhou Crucial role of heme oxygenase-1 in the sensitivity of acute myeloid leukemia cell line Kasumi-1 to ursolic acid. Anticancer Drugs. 2014 Apr;25(4):406-14
The authors show that YD15 cells were sensitive to TW-37 treatment while MC3 cells were not. This was, among others shown using a cell viability assay of cells treated with 0 µM and 1.25 µM TW-37. Because the response of different cell lines towards specific inhibitors can be highly heterogeneous, it is possible, that higher concentrations are required for MC3 cells. Please show cell viability and cell morphology (conventional light microscopy) using a wide range of TW-37 concentrations.
☞ As you commented, we determined cell viability and cell morphology using a wide range of TW-37 concentrations. The results showed that higher concentrations of TW-37 can inhibit the viability of both MC3 and YD-15 cells and MC3 cells are more insensitive to TW-37 than YD-15 cells.
The flow cytometry profiles shown in Figure 1D suggest that the TW-37-treated MC3 cells were not gated correctly.
☞ As you commented, we re-gated them. The below is our correction. We added new Fig. 1. [line 91]
Please show mRNA levels of HO-1 and HSP27 after overexpression using qRT-PCR.
☞ Thank you for your kind comments. However, we don’t think we need to show mRNA levels of HO-1 and HSP27 after overexpression because we already showed the protein levels of HO-1 and HSP27 using western blotting. Thank you for your comments again.
The H2DCF-DA staining (Figure 4A) seems to be unspecific (high background staining). Please provide pictures with a higher magnification which make clear that the staining was specific.
☞ As you suggested, we provide pictures with a higher magnification to make it clear that the staining was specific. We added new Fig. 4. [line 164]
Please indicate for all experiments whether only adherent or also floating cells were collected.
☞ We collected both adherent and floating cells for all experiments. We described it in ‘Materials and Methods section.’ [line 272]
Cleaved PARP is not expressed (suggesting that it results from gene transcription) as mentioned in the text (line 95) but results after cleavage of the PARP protein by active caspases. Please avoid the term “expression” in this context.
☞ As you suggested, we corrected it like the below. [lines 79, 81, 95 and 157]
‘The expression of cleaved PARP à cleavage of PARP’
Please add scale bars to all microscopy images.
☞ As you commented, we added scale bars for all microscopy images.
Please italicize “in vivo” and “in vitro”.
☞ As you commented, we italicize “in vivo” and “in vitro”. [lines 51, 61 and 73]
Human gene names should be written in capital letters and be italicized. Human proteins are written in capital letters. Murine gene names should be written in small letters (only the first letter is capitalized) and be italicized. Murine proteins are written in capital letters.
☞ As you suggested, we corrected it.
